# Soil Aggregate Breakdown with Colloidal Particles Release and Transport in Soil: A Perspective from Column Experiments

Gang Cao [1], Bokun Chang [1], Zhiying Zhou [1], Liang Hu [1], Wei Du [1,2] and Jialong Lv [1,2,*]

1    College of Natural Resources and Environment, Northwest A&F University, Yangling 712100, China
2    Key Laboratory of Plant Nutrition and the Agri-Environment in Northwest China, Ministry of Agriculture, Yangling 712100, China
*    Correspondence: ljlll@nwafu.edu.cn

**Abstract:** The strongest fortresses often disintegrate from the inside. Likewise, soil internal forces play a critical role in the initial breakdown process of soil aggregate, thus accelerating soil erosion and the release of soil colloid particles. To date, research on the effect of soil internal forces, especially separating the electrostatic force, and on the process of soil aggregate breakdown with particle release and transport in soil is largely inadequate. Therefore, column experiments were used to investigate the properties of transport and soil particles released from the disintegration of model soil aggregates caused by different levels of electrostatic forces. We found that the increase of electrostatic repulsive pressure was the immediate cause of soil aggregate breakdown, that the highest concentration of released soil particles could reach 808.36 mg L$^{-1}$, and that the mean particle sizes of the released soil ranged from 100 nm to 300 nm. The particle size distributions and clay mineral composition of the released soil particles were not dominated by the electrostatic force. In practice, the change of external conditions of agricultural soil would lead to the change of soil internal forces, then affect soil aggregate stability. This study aims to provide a micro perspective to understand the release of fine particles from soil matrix and its implication for agricultural soil.

**Keywords:** aggregate stability; colloidal particle releasing; column experiments; electrostatic repulsive force; particle properties





## 1. Introduction

As the basic unit of soil structure, soil aggregates and their stability are closely concerned with the questions of various soil processes and environmental problems. For instance, soil aggregates influence the transport of water, air, and heat in soil, support the soil structure to resist soil erosion, and control microbial activity, thus affecting migration and transformation of matter and energy in soil [1–4]. The degeneration of soil structure and soil erosion caused by soil aggregate breakdown threaten agricultural and environmental safety [5]. Therefore, it is of significance to investigate the internal mechanism of soil aggregate stability.

It is widely accepted that the main mechanisms of soil aggregate breakdown can be divided into slacking, physicochemical dispersion, differential swelling, and raindrop impact [6,7]. Although the change of the external conditions (precipitation, erosion, solution chemistry mobilization, etc.) usually leads to the release and transport of soil colloid particles in soil, researchers realize that the essence of soil aggregate breakdown is highly related to the soil internal forces [8–11]. Soil internal forces, i.e., the interactions among soil particles on the mesoscopic scale, including DLVO force (van der Waals force and electrostatic force) and non-DLVO force (e.g., hydration repulsive force), significantly influence a series of soil physical and chemical processes [6,10]. In particular, the aggregation phenomena that happened in soil largely depend on the DLVO force [12–14], and the hydration repulsive force significantly contributes to the initial breakdown process of the

"dry" aggregates [10,11]. In the raindrop simulation experiments [15], the contribution rate of soil internal forces was more than the external impact in splash erosion under dilute solutions. Essentially, soil internal forces influence soil water movement through water matric potential attributed to soil aggregate disintegration and soil pore morphology [16]. Despite the mechanism of the internal forces on soil aggregate stability being well investigated, the current experimental methods hardly separated the effect of the electrostatic force from them.

The fine soil particles, an important source of soil electrochemical properties, are usually released with soil aggregate breakdown. According to the estimation [17], the colloidal-sized particles (<1 μm) contribute more than 80% of the surface charge and the specific surface area of the whole soil. On the other hand, because of the strong mobility and adsorptive capacity of colloidal particles, there has been an increasing concern about the risks, such as soil pollution, soil degradation, or water eutrophication, caused by soil colloid particles that facilitated the transport of contaminants or nutrients [18–23].

In field research, it is relatively difficult to research the properties and transport behaviors of soil particles within the limitation of distance and time periods [24]. Therefore, a column experiment is the most common and efficient method used in laboratory research. Compared with the traditional methods applied to soil aggregate stability, e.g., wet sieving, ultrasonic vibration, pipette method, rainfall simulation, etc., the column experiment has advantages in the investigation of particle release and transport, which could support tracking the released particles directly influenced by the internal forces. The used column experiments, although making great progress during the past few decades in describing and predicting the transport or co-transport of the target objectives, predominantly rely on physically and chemically stable model porous media (e.g., glass beads, quartz sand, fractured rocks, etc.) [25–27] or on the stabilized soil aggregates [28]. Thus, it is helpful to extend the application range of the column experiment by knowing the properties and transport of the mobilized colloidal particles released from soil aggregate breakdown.

It is essential to understand the role of the internal forces in the breakdown process of soil aggregates and build up rational cognition on the consequential particle release and transport in soil. In this research, column experiments were used to control the variables and separate the effect of the electrostatic repulsive force from the whole internal forces. The specific objectives of this study were: (1) to investigate the releasing properties of soil particles from aggregate breakdown in soil; (2) to clarify the mechanism of the electrostatic repulsive force on soil aggregate breakdown; and (3) to determine the properties and composition of the released soil colloids under electrostatic force.

## 2. Materials and Methods

### 2.1. Soil Collection and Pre-Treatment

Lou soil was collected in Yangling (108°5′11″ E, 34°17′54″ N), Shaanxi Province, China (Figure S1). Lou soil is a representative soil in the semi-arid area under long-term cultivation. Winter wheat (*Triticum aestivum Linn*) and maize (*Zea mays* L.) are the staple crops in this region. According to the FAO soil classification, the Lou soil that developed from loess parent materials was classified as Calcic Cambisols. The texture was silty loam (18.28% clay, 55.09% silt, and 26.63% sand) based on the International Classification System of Soil Texture.

Pre-treatment was needed for the collected soil to obtain a purified and homogeneous surface so that the soil internal forces could be quantitatively calculated [10]. In brief, air-dried soil (2 kg) and 0.5 mol $L^{-1}$ KCl (10 L) solution were added into a clean barrel and then stirred for 24 h. The suspension was allowed to stand for 12 h and removed the supernatant. This process was repeated 3 times. Then, deionized water (10 L) was added in and stirred for 24 h. The mixture was centrifuged at 5000 rpm for 5 min to remove the surplus $K^+$. After 2 times, the centrifuged soil was dried at 60 °C in an oven and then sieved. The 0.5–1 mm model soil aggregates were stored for the next experiments.

## 2.2. Soil Analytics

The basic properties of Lou soil, particle size distributions, pH (water: soil = 2.5:1), and organic matter content, were analyzed using the laser diffraction device (Mastersizer 2000, Malvern Instruments Ltd., Malvern, UK), the laboratory pH meter (FE 20, MET-TLER TOLEDO Instrument Shanghai Co., Shanghai, China), and the potassium dichromate oxidation-external heating method, respectively. The electrochemical properties and specific surface area (SSA) were obtained using the combined determination methods based on ion exchange equilibrium (details in Text S1) [29,30]. The morphology, elemental composition, and clay mineral composition of the packed soil aggregates and the collected soil particles in the column experiment were measured by field emission scanning electron microscope, energy dispersive spectrometer (S-4800, Hitachi, Ltd., Tokyo, Japan), and X-ray diffraction (D8 ADVANCE A25, Bruker Co., Ltd., Karlsruhe, Germany), respectively.

## 2.3. Soil Column Experiments

Cylinder plexiglass columns—10 cm in length and 1.5 cm inner diameter—were used in the experiments. The experimental arrangements were the same as our previous column experiment (Figure S2) [31]. The $K^+$-saturated soil aggregates were carefully dry-packed into the columns, and approximately 16 g soil aggregates were added. Each column was carefully packed with gentle vibration, and the nylon nets were placed at the top and bottom of the column to support the aggregates. KCl solution ($10^{-1}$ M, pH 7) was introduced from bottom to top into the soil column at the flow velocity of 0.1 mL min$^{-1}$ using a peristaltic pump (HL-2B, Shanghai Huxi Analytical Instrument Factory Co., Ltd., Shanghai, China), because a high concentration of electrolyte maintained the stability of soil aggregates [16]. After water-saturating the soil column, the flow rate was gradually increased to 0.75 mL min$^{-1}$ for a half hour and then adjusted to 0.5 mL min$^{-1}$ and maintained for 1 h to steady the hydrologic environment. The next experiments were carried out in two phases: Phase 1, 20 PVs (pore volume) of electrolyte solutions with different concentrations ($10^{-5}$, $10^{-3}$, $10^{-2}$, $10^{-1}$, and 1 M) was introduced; Phase 2, 2.5 PVs of $10^{-1}$ M KCl solution was injected. The effluents with released soil particles were collected every 10 min using a partial collector (EBS-20, Shanghai Huxi Analytical Instrument Factory Co., Ltd., Shanghai, China). Two replicates were performed for each run. The concentrations and the particle size distribution of soil particles were quantitatively determined using an ultraviolet-visible spectrophotometer (UV-2600, Shimadzu Co., Ltd., Japan) at 700 nm and dynamic light scattering (NanoBrook 90Plus Zeta, Brookhaven Instruments Corporation, Brookhaven, USA), respectively. Tracer experiments were launched to determine the hydrodynamic conditions of the soil columns (Text S2). The calibration curve of the soil particles is shown in Figure S3 ($R^2$ = 0.999).

## 2.4. Mathematical Model

The interaction forces between soil particles can be divided into repulsive pressure and attractive pressure [15]. Here, the attractive van der Waals pressure ($P_{vdw}$) can be calculated using Equation (1):

$$P_{vdw} = -\frac{A}{0.6\pi}(10d)^{-3} \tag{1}$$

where A (J) is the effective Hamaker constant of Lou soil, which is $4.54 \times 10^{-20}$ J in this study.

The repulsive pressure can be calculated using Equations (2) and (3) [11,32]:

$$P_E = \frac{2}{101}RTc_0\left\{\cos h\left[\frac{ZF\varphi_{(\frac{d}{2})}}{RT}\right] - 1\right\} \tag{2}$$

$$P_h = 3.33 \times 10^4 e^{-5.76 \times 10^9 d} \tag{3}$$

where $P_E$ and $P_h$ are electrostatic pressure and hydration pressure of soil particles, respectively, R (J mol$^{-1}$ K$^{-1}$) is gas constant, T (K) is the absolute temperature, $c_0$ (mol L$^{-1}$) is the equilibrium concentration of the cation in the bulk solution, Z is the valence of cation, F (C mol$^{-1}$) is Faraday's constant, $\varphi_{(d/2)}$ (V) is the potential at the middle of the overlapping position of the electric double layers of two adjacent particles, and d (dm) is the distance between two adjacent particles.

In Equation (2), the $\varphi_{(d/2)}$ can be calculated using Equation (4) [33]:

$$\frac{\pi}{2}\left[1+\left(\frac{1}{2}\right)^2 e^{\frac{2ZF\varphi_{(\frac{d}{2})}}{RT}} + \left(\frac{3}{8}\right)^2 e^{\frac{4ZF\varphi_{(\frac{d}{2})}}{RT}}\right] - \text{arcsine}\, \frac{ZF\varphi_0 - ZF\varphi_{(\frac{d}{2})}}{2RT} = \frac{1}{4}d\kappa e^{\frac{-ZF\varphi_{(\frac{d}{2})}}{2RT}}$$

(4)

where $\varphi_0$ (V) is the surface potential, $\kappa$ (dm$^{-1}$) is Debye–Hückel parameter, and $1/\kappa$ is the thickness of the EDL (electric double layer). The 1:1 type electrolyte used in the experiments was KCl; therefore, they can be calculated using Equations (5)–(7):

$$\varphi_0 = -\frac{2RT}{ZF}\ln\left(\frac{1-a}{1+a}\right).$$

(5)

$$\frac{\kappa CEC}{SSA_{C0}} = 1 + \frac{4}{1+a} - \frac{4}{1+e^{-1}a}$$

(6)

$$\kappa = \left(\frac{8\pi F^2 Z^2 c_0}{\varepsilon RT}\right)^{\frac{1}{2}}$$

(7)

where $a$ is the intermediate variable, $CEC$ (mol g$^{-1}$) is the cation exchange capacity, $SSA$ (m$^2$ g$^{-1}$) is the specific surface area, and $\varepsilon$ (F m$^{-1}$) is the dielectric constant of water.

We defined the column index (CI) to rate the mass loss of the packed soil and normalize the results in different experimental condition:

$$CI = \frac{\int_0^{t_0} c_t dt}{M_0}$$

(8)

where $t_0$ (min) is the whole experimental time, $c_t$ is the concentration of soil particle at $t$, $CI$ (mg g$^{-1}$) is the particle loss per gram of the soil aggregates, and $M_0$ is the mass of total packed soil aggregates in a soil column. In this manuscript, accumulation of soil particles was still used for analysis.

The hydraulic parameter dispersion coefficient (D) was simulated through the results of a tracer experiment using the Advection Dispersion Model (Text S3).

## 3. Results and Discussion

### 3.1. Soil Properties

The pH and organic matters of the Lou soil were 8.21 and 5.30 g kg$^{-1}$, respectively. The cation exchange capacity (CEC) was 21.73 cmol kg$^{-1}$, and specific surface area (SSA) was 42.36 m$^2$ g$^{-1}$. The surface potentials of the soil and EDL thicknesses of a single particle under different KCl concentrations are listed in Table 1. With a decrease in the background electrolyte concentration, the surface potentials (absolute value) and EDL thickness were increased. The semiquantitative results of XRD showed that clay minerals of used Lou soil mainly consisted of illite, vermiculite, kaolinite, and montmorillonite. From the SEM image, a single soil aggregate was composed of a large number of fine clay minerals. The EDS results demonstrated that the dominant elements Si, O, and Al were consistent with the elementary composition of clay minerals from XRD. Moreover, the fitted dispersion

coefficient value (*D*) of the soil columns at a velocity of 0.5 mL min$^{-1}$ was 0.46 cm$^2$ min$^{-1}$ (Figure S4).

**Table 1.** Surface potential and EDL thickness of the soil under different KCl concentrations.

| KCl Concentration | Surface Potential | EDL Thickness |
|---|---|---|
| (mol L$^{-1}$) | (mV) | (nm) |
| 10$^{-5}$ | −404.43 | 971.04 |
| 10$^{-3}$ | −286.61 | 97.10 |
| 10$^{-2}$ | −228.17 | 30.71 |
| 10$^{-1}$ | −170.97 | 9.71 |
| 1 | −116.65 | 3.07 |

### 3.2. Releasing, Transport, and Accumulation of the Disaggregated Soil Particles

The direct consequence of soil aggregate breakdown was the release and transport of soil particles. As manifested in Figure 1, it can be easily concluded that the change of the background electrolyte concentrations significantly influenced the soil particle concentrations. When the concentrations of the background electrolyte remained or increased to 1 M, soil particles were hardly detected in the whole process. Note that soil particles were also not observed in the stabilization stage where the flow velocity was 0.75 mL min$^{-1}$. In contrast, the packed soil aggregates were broken and released particles when the concentrations of KCl were 10$^{-2}$ M or less.

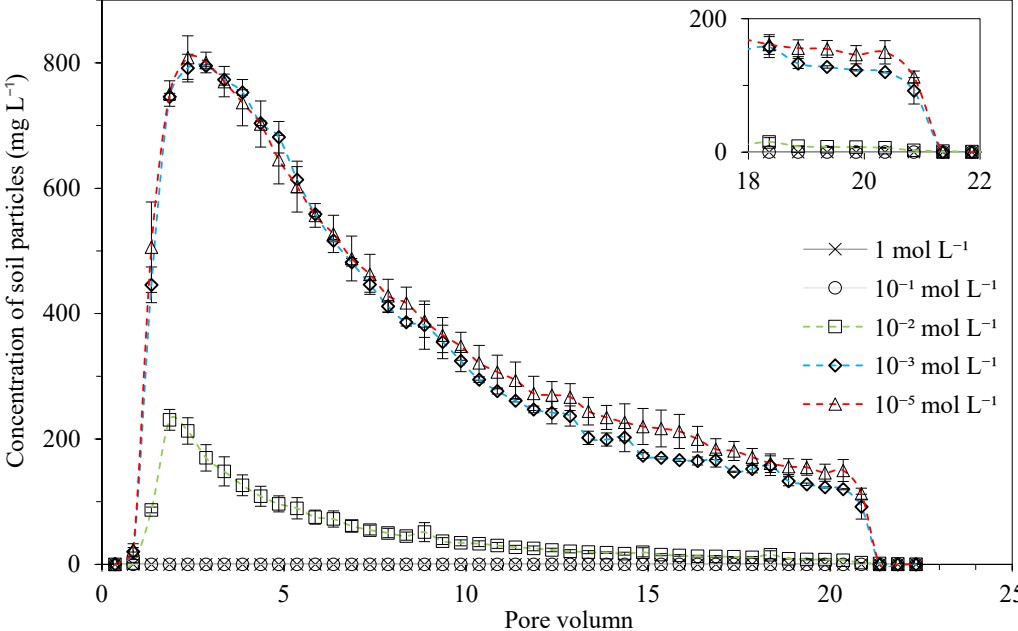

**Figure 1.** Effluent concentrations of soil particles under different electrolyte concentrations.

The concentrations of the released soil particles rapidly increased and then decreased. For the conditions of 10$^{-5}$, 10$^{-3}$, and 10$^{-2}$ M, the particles coincided with the arrival of the infiltration front at the end of the first PV, where the concentrations were starting to rise. The phenomenon, as observed before, was due to the sensitive and responsive reaction of aggregates meeting the low concentration of electrolyte [14]. Furthermore, the earlier breakthrough of the soil particles was attributed to part of the preferential flow generated by macro-pores [34]. Those two factors also contributed to the delay of the maximum concentration of soil particles at low KCl concentrations (10$^{-3}$ and 10$^{-5}$ M) compared to 10$^{-2}$ M. In the second PV when the background electrolyte solution was almost replaced by 10$^{-2}$ M KCl, the concentration of the soil particles reached the highest

point of 230.51 mg L$^{-1}$. In the comparison of $10^{-5}$ M and $10^{-3}$ M, the highest points, which were delayed and maintained for a PV, reached 808.36 mg L$^{-1}$ and 795.28 mg L$^{-1}$, respectively. Then, there was a progressive decline in the concentration of soil particles after reaching the peak value. The difference between the curves of $10^{-5}$ M and $10^{-3}$ M was not significant until the end of the sixth PV, while the soil particle concentrations of $10^{-5}$ M were higher than that of $10^{-3}$ M in the rest, in part due to the response of the lower KCl concentration.

With the reinjection of $10^{-1}$ M background electrolyte solution, the breakdown process of soil aggregates was suddenly halted. It was at the end of the 21st PV that the concentrations of soil particles significantly decreased. The concentration variation rate of released soil particles (Figure S5) at $10^{-5}$ M and $10^{-3}$ M reached the minimum value of −112.75 and −91.95 mg L$^{-1}$ per half PV, respectively. It should be pointed out that the minimum variation rate was actually caused by the effect of $10^{-1}$ M KCl. The same but converse trend manifested in the injection of low concentrations (below $10^{-1}$ M) of electrolyte, such as the maximum velocity change rate at 1.5–2 PV rapidly decreasing until the minimum. The change of the concentration of the background electrolyte solution was a crucial factor that affects the soil aggregate stability.

For the whole particle releasing curves, the reduction of concentrations was partially irregular, and exception values were partly influenced by the porosity of the soil columns and heterogeneity of the soil aggregate distribution [34,35]. As known, a single sea wave is unpredictable, but the direction of the whole tide is definite. Due to the purified and homogeneous surface of the treated model aggregates, we could consider that all the soil aggregates contributed to the release of soil particles rather than being overly concerned with the asynchronous process of an individual aggregate breakdown. With the stabilizing of solution chemistry, the effluent concentrations were also relatively stabilized. A similar decline trend was also observed in the long-term colloid particle release experiments from natural soil [36], and the long tailing in the particle-releasing curves indicated there was a limited but durable supply of aggregates contributing to a slow and long-term release of soil particles [37].

Figure 2 demonstrates the mass quantity accumulation of the disaggregated soil particles. The accumulated loss of soil particles increased rapidly at first and then slowly increased to the end. The accumulation curves could be well fitted using logarithmic equations, and the fitting coefficients ($R^2$) were 0.99, 0.99, and 0.98 with the increase of KCl concentrations from $10^{-5}$ to $10^{-2}$ M. The fitted equations demonstrated a proper perspective to understand the properties of particle release and were useful to rebuild and estimate the results of soil aggregate breakdown. The rainfall simulation used to rate the stability of soil aggregates [38] also showed the same trend in particle release. The *CI* presented the degree of soil aggregates breakdown, and the values were 4.70, 4.45, and 0.67 mg g$^{-1}$ from $10^{-5}$ to $10^{-2}$ M. The minimum *CI* from the KCl concentration of $10^{-2}$ M was much smaller than the rest. By contrast, the values were relatively close for $10^{-5}$ and $10^{-3}$ M. Here, we introduced the concept of critical concentration. When the electrolyte concentration was less than the critical value, the disaggregation process was accelerated; otherwise, it was stopped. The critical concentration of $10^{-2}$ M in soil aggregate stability has been confirmed in many reports [39,40]. The results from Lægdsmand et al. [41] proved the relationship between the released colloid concentrations and the reciprocal of the square root of EC (electrical conductivity as an indicator of electrolyte).

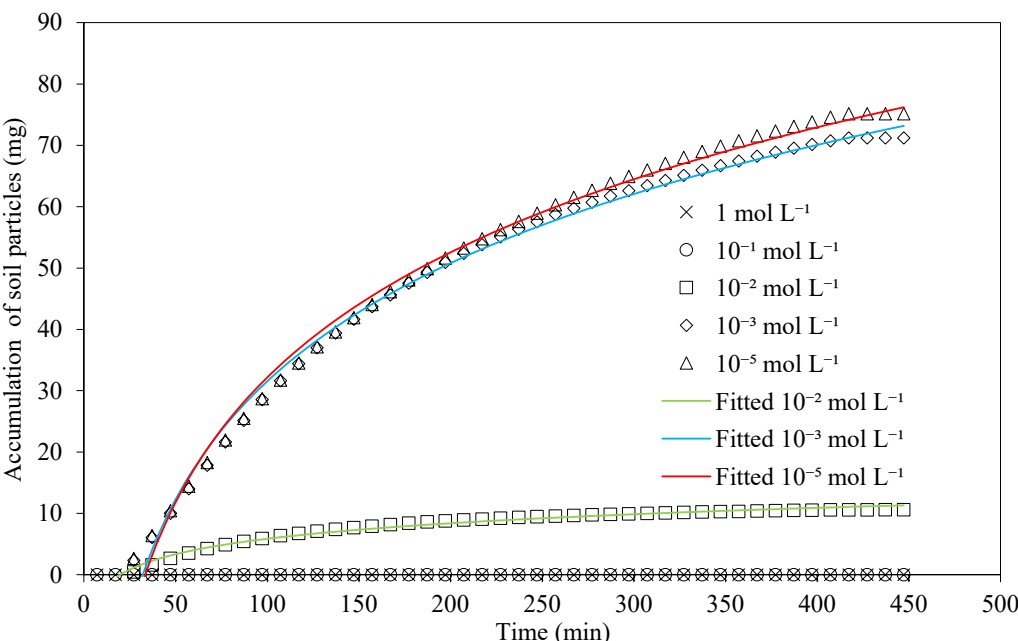

**Figure 2.** Accumulation of released soil particles under different electrolyte concentrations with time.

Apparently, the column experiments in this study separated the influence of physicochemical dispersion from the four main mechanisms of soil aggregate breakdown well. The saturated soil surface with homogeneous cations simplified the experimental conditions, and the water-saturated process with high electrolyte concentration prevented the influence of the slacking effect caused by the compression of ambient air, clay mineral differential swelling, and raindrop impact. The only variable was the background electrolyte concentration, and $10^{-1}$ M KCl maintained the stability of soil aggregates and terminated the release of soil particles. The physicochemical dispersion is essentially the effect of soil internal forces. Furthermore, the change of soil internal force related to experimental conditions is revealed and discussed in the following section.

### 3.3. Dependence of Transport of Disaggregated Soil Particles on Electrostatic Repulsive Force

The DLVO (Derjaguin, Landau, Verwey, and Overbeek) theory, which is successful in quantitative interpretation of particle stability, has been expanded to environmental and engineering fields and is still making substantial progress [16,42–45]. Hu [11] and Xu [14] have shown excellent images about the effect of internal forces (electrostatic force and hydration force) on the swelling and explosion process of soil and clay aggregates under low electrolyte concentrations. However, there is still confusion about which one plays the critical role in aggregate breakdown.

In our column experiments, the pre-saturated process with a high concentration of background electrolyte ($10^{-1}$ M) significantly decreased the total repulsive force. In particular, there was hardly any release of soil particles in the random checks of the effluents during the process. The potential external influences, such as air pressure, direct impact, etc., were well excluded. Compared to the pipette method which was used to rate the stability of soil aggregates [39], the column experiments did not introduce extra uncontrollable force from the operator and avoided the influence of the existing air in the cylinder which might magnify the slacking effect at the air–water interface during the experimental process. The two factors mentioned above might magnify the degree of aggregate breakdown and the amount of released soil colloid particles. Therefore, our experiments might be more appropriate to quantitively estimate aggregate stability under the effect of soil internal forces. Additionally, the gravity effect was insignificant on the mobilization and transport of the fine soil particle in the column experiments [46].

The electrostatic repulsive pressure originated due to the existed charges at the soil particle surface [43]. The decline of background electrolyte concentration weakened the shielding effect of cations on the electric field around the soil particle and varied the partial ion concentration relative to the bulk solution, thus driving ion movement under the concentration gradient and potential gradient and resulting in the redistribution of the EDL [47,48]. This led to the overlap of the previously untouched EDL, which directly and quickly strengthened the effect of the electrostatic repulsive pressure between the two adjacent soil particles. After calculation, the surface potential and EDL thickness of the soil particle were $-170.97$ mV and 0.85 nm (Table 1) after stabilizing the physical and chemical conditions of the soil columns. When the background electrolyte concentrations were less than $10^{-1}$ M in this study, the surface potentials (absolute values) and EDL thicknesses of the soil particle were significantly increased. For example, the surface potential increased to $-286.61$ mV, and the thickness of EDL increased tenfold, which provided favorable conditions for the breakdown of the soil aggregates.

The quantitative calculation could rate the soil internal forces well. The distributions of the electrostatic repulsive pressure under different KCl concentrations are shown in Figure 3a. The electrostatic repulsive pressure declined with the increase of the electrolyte concentration and distance between two adjacent particles. At the same distance, the lower the electrolyte concentration, the greater the electrostatic repulsive pressure. The distributions of the concentration-independent surface hydration repulsive pressure, van der Waals attractive pressure, and its resultant are manifested in Figure 3b. Different from electrostatic pressure, those two pressures were distributed at a relatively short distance (hydration pressure less than 2 nm), which manifested as the coincidence of its resultant and van der Waals pressure. The distribution of net pressure (summary of electrostatic repulsive force, van der Waals force, and surface hydration force) shown in Figure 3c indicated that the rapidly decaying electrostatic repulsive pressure at electrolyte concentration of 1 and $10^{-1}$ M led to a decrease of the net pressure, thus stabilized the soil aggregates in aqueous solution and that net pressures were repulsive at any distance when KCl concentration was lower than the critical concentration, $10^{-2}$ M. The results were the situation of $10^{-2}$–$10^{-5}$ M when the disaggregation of soil aggregates happened.

The role of electrostatic repulsive pressure which directly contributed to the change of the soil internal force in the transport of soil particles was revealed. Figure 4 shows the relationship of the net pressure ($10^{-2}$, $10^{-3}$, and $10^{-5}$ M) at different distances (1.5, 2, 4, 6, 8, and 9 nm) with the accumulation of soil particles (i.e., *CI*). More detailed parameters are shown in Table S2. As shown in Figure 4, there was a linear relationship between them at given distances. The correlation coefficients ($R^2$) ranged from 0.996 to 1.000. For instance, the distance of 6 nm represented soil aggregates that were stable in the columns because the net pressure of $10^{-1}$ M was about to be negative. The corresponding net pressures were 1.84, 2.24, and 2.29 atm from $10^{-2}$ M to $10^{-5}$ M and $R^2 = 0.998$. In addition, the fitted result simulated the relationship between the net pressure and the accumulation of the released soil particles at the distance of 2 nm well. When the distances were less than 2 nm, the fitted results were relatively worse. It might indicate that the effect of hydration repulsive pressure in the column experiment might be much less than that of the electrostatic repulsive pressure.

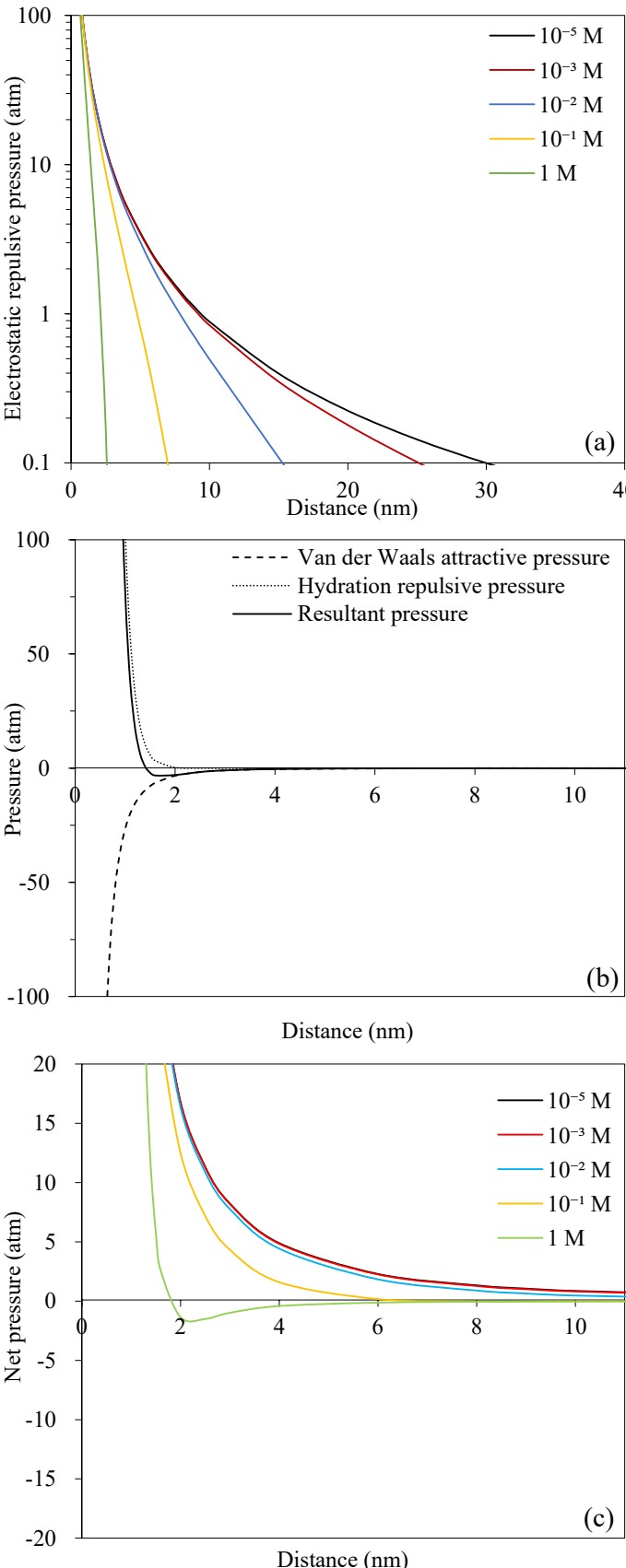

**Figure 3.** Distributions of electrostatic repulsive pressure(**a**); hydration repulsive pressure, van der Waals attractive pressure, and its resultant pressure (**b**); net interaction pressure (**c**).

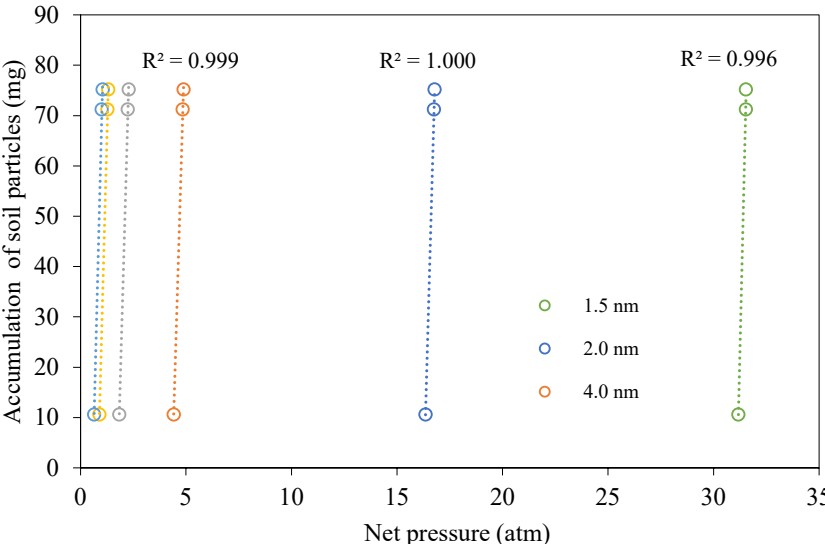

**Figure 4.** Relationship of net pressure and accumulation of soil particles.

On the other hand, a single soil aggregate was surrounded by other aggregates. The pressure from the surrounding aggregates limited the explosion of a single aggregate. From a macro perspective, though the disintegration of soil aggregates will not change the appearance of soil, it provides favorable conditions for rill erosion and accelerates the erosion process under the influence of the runoff [49].

### 3.4. Characteristics of Particle Size and Mineral Composition of Released Soil Particles

We already concluded that the releasing properties of soil particles were dominated by the electrostatic repulsive pressure. The further question is whether the electrostatic pressure influenced the other properties of the released colloidal particles.

The particle size distributions of the effluents were analyzed. Figure 5a illustrates the average size of the released particles with pore volume from 2 to 21. It should be noted that the largest sizes of soil particles were much smaller than that of the nylon net mesh. During the whole disaggregation process, different electrolyte concentrations of $10^{-5}$, $10^{-3}$, and $10^{-2}$ M showed the same distribution on the average particle size of the discharged soil particles which mostly ranged from 100 nm to 200 nm. After a Kruskal–Wallis H Test (R version 3.5.3), the results showed there were no statistical differences ($p = 0.27 > 0.05$) in the average particle size among $10^{-5}$, $10^{-3}$, and $10^{-2}$ M KCl (Figure S6). That is to say, the discrepancies of the repulsive electrostatic pressure were not the critical factor that influenced the average disaggregated soil particle size. The leaching experiments [41] also showed the size distributions of the released colloid did not vary with the experimental process. Limited by the measurement method (single-particle counter), Lægdsmand only detected the colloid particle >100 nm. Here, we confirm that the undetected part of the particles took a nonnegligible part of the whole discharged particles. The recent colloid generation experiments from compacted bentonite indicated that the size of the released particles ranged from 50 to 100 nm both in dynamic and static conditions [50].

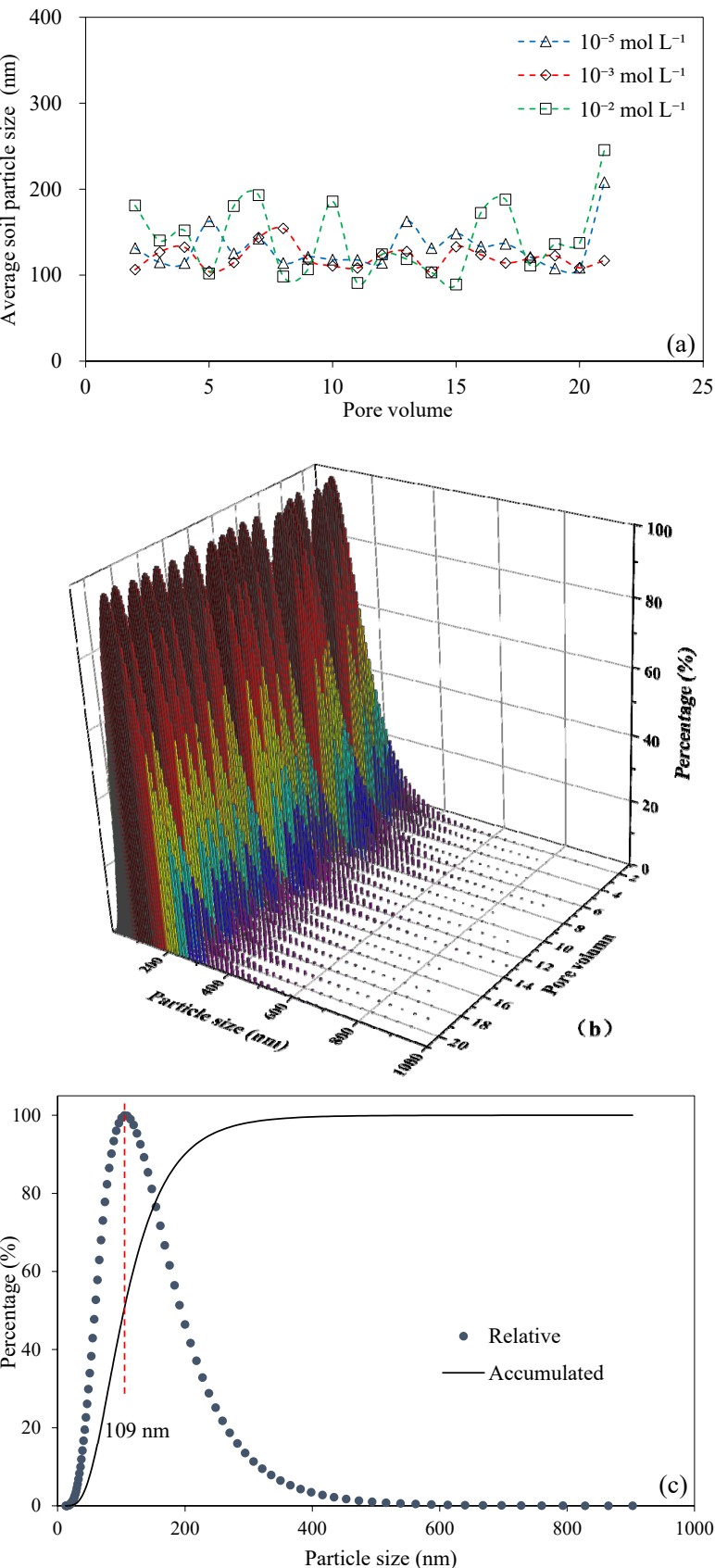

**Figure 5.** Average particle size variation of released particles with pore volume (**a**); the whole particle size distribution of the effluents under $10^{-3}$ M (**b**); particle size distribution at the 2nd PV (**c**).

Due to the similar particle size distributions, we only show the situation of $10^{-3}$ M in Figure 5b. Both Figure 5b,c show that a fraction of the released soil particles was larger than 200 nm (<10%), and that the maximum particle size could reach 1 μm. Overall, the ranges of soil particle size were very wide, and the mean particle size was only a characterization of the whole particles. The distributions of the mean size of particles were therefore stochastic and irregular. On the other hand, the random and asynchronous disaggregation process led to the differential distribution of soil particle sizes.

The release, filtration, and deposition also affected the transport of the disaggregated particles [51]. For an individual particle, the repulsive pressures dominated the interaction with other particles. However, at the same time, the negatively charged medium surface hardly provided available deposition sites. Therefore, the effect of deposition was limited. Physically, the straining process of soil particles could happen when the size of the particles was larger than the pore throats. The value 0.0017, a ratio of particle diameter to medium diameter, was commonly chosen as the critical straining threshold to approximately rate the influence of straining [52]. Here, we took the minimum size of the packed soil aggregates of 0.50 mm for estimation. The computed result illustrated that the maximum particle size which could pass through the soil column was 850 nm. Considering a minute quantity of large particles was detected, the very release process might play a certain role in the size of released particles. The reason might be that the interactions of the surrounding large particles stabilized the particle which maintained the overall structure of soil aggregates. Therefore, the particle size of released particles was largely attributed to the filtration and release process.

There was no evidence that the electrostatic repulsive pressure had a significant separation or diversion effect on the clay mineral composition of the disaggregated soil particles. As observed, there was hardly any micro aggregate, but there were individual particles in the effluents. The observation was consistent with the previous study [53]. After collecting and oven drying the disaggregated soil particles, the comparison of released particles and packed aggregates was demonstrated in Figure 6. Although there were obvious differences in morphology, both soils were similar in elemental composition. The most abundant elements (Si, O, Al) accounted for more than 70% of the total, which was also the main elements of clay minerals. Similarly, the XRD results in Figure 6d indicate the clay minerals of the two were illite, vermiculite, kaolinite, and montmorillonite in descending order by content. It can be confirmed that all the soil particles released from the breakdown of aggregates were soil elementary particles, and the mineral composition of released particles was highly correlated to the raw soil [7].

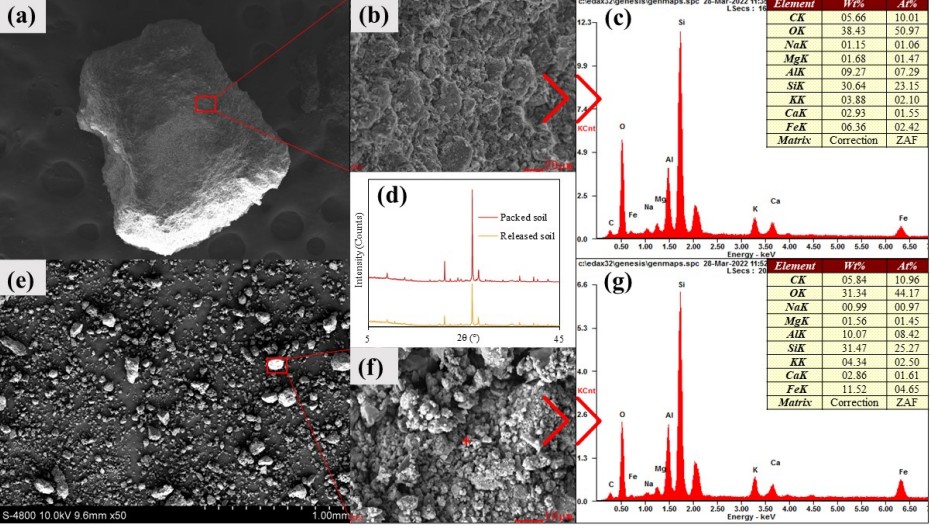

**Figure 6.** SEM images, EDS, and XRD (**d**) of packed soil aggregate (**a**–**c**) and released soil particles (**e**–**g**) (rectangles in (**a**,**e**) do not represent the true scales).

The soil particles discharged from aggregates presented high mobility. A previous study launched in the saturated porous media proved that the small size soil particles (<1 μm) have stronger mobility than the large (10–1 μm) [54]. Specifically, almost all the discharged soil particles were less than 1 μm in the breakdown process.

*3.5. Implications*

"Where do these ubiquitous colloidal particles in the environment come from"? Our results indicated that the soil was the source and sink of soil colloid particles. The change of external conditions (precipitation, snowmelt, irrigation, etc.) and disturbance of groundwater conditions accounted for the release and stabilization of the colloidal particles through soil internal forces [27,55,56]. It is the porous media that could release or mobilize these fine particles, such as the report that the colloids in groundwater mainly originated from the upper farmlands [37,46].

The initial breakdown of the aggregates contributes to the release of soil colloid particles and the exposure of soil organic matter. The former could result in agricultural and ecological problems (e.g., soil fertility decline, soil erosion, farmland non-point source pollution, water eutrophication, etc.) through facilitated transport of nutrients or pollutants by soil colloid, and the latter is closely related to the greenhouse effect due to emission of greenhouse gases ($CO_2$, $CH_4$, $N_2O$, etc.) of the farmland soil from accelerating decomposition of exposed organic substance [3–5,7,19,57–59]. Finally, with the sedimentation of the released fine particles, the soil crust formed and resulted in the reduction of water infiltration and loss of soil electrochemical properties [5,17]. These results could lead to the degradation of soil structure.

## 4. Conclusions

The column experiments have the potential to be a standard method to estimate soil aggregate stability. The influence of the electrostatic repulsive force was well separated. The release and properties of the discharged soil colloidal particles were quantitatively determined and investigated. The results show that the electrostatic repulsive pressure was the immediate factor in the initial breakdown process of soil aggregates. The concentrations of released colloidal particles increased rapidly and then decreased. Furthermore, the accumulation of released particles has a linear relationship to the net pressure which was largely affected by the electrostatic repulsive pressure. However, soil internal forces did not have significant effects on the particle size distributions and clay mineral composition. The released soil particles had average sizes ranging from 100 nm to 300 nm. The breakdown process, filtration, and deposition dominated the transport of released particles. We should be aware of the negative implications. This study can be a supplementary explanation of the breakdown mechanism of soil aggregates. In practice, future column experiments based on soil media should consider the release and the facilitated transport effect of soil colloid particles from the matrix. Additionally, the natural environment is much more complex than the surface saturated model soil in the laboratory. The soil organic matter which is significant in soil stability should be taken into account in future investigations.

**Supplementary Materials:** The following supporting information can be downloaded at: https://www.mdpi.com/article/10.3390/agriculture12122155/s1, Text S1: The combined determination method; Text S2: Tracer experiments; Text S3: Advection Dispersion Model; Table S1: Fitted parameters of accumulation curves; Table S2: Fitted parameters of accumulation curves; Figure S1: Map of sampling site; Figure S2: Experimental arrangements; Figure S3: Calibration curve of the Lou soil particles; Figure S4: Simulated results of the tracer transport; Figure S5: Concentration variation rate of released soil particle; Figure S6: Violin plot of mean particle size of released soil; Figure S7: Particle size distribution of soil particle with pore volume under $10^{-5}$ M (a) and $10^{-2}$ M (b).

**Author Contributions:** Conceptualization: W.D.; data curation and methodology: B.C.; funding acquisition: W.D. and J.L.; investigation: G.C.; resources and software: Z.Z.; supervision: J.L.;

validation: L.H.; writing—original draft: G.C.; writing—reviewing and editing: G.C. and W.D. All authors have read and agreed to the published version of the manuscript.

**Funding:** This research was supported by the National Natural Science Foundation of China (Nos. 42107332 and 42077135), and the Natural Science Basic Research Program of Shaanxi Province, China (No. 2021JQ-170).

**Institutional Review Board Statement:** Not applicable.

**Data Availability Statement:** The data presented in this study are available on request from the corresponding author.

**Acknowledgments:** The authors wish to thank the National Natural Science Foundation of China (Nos. 42107332 and 42077135), and the Natural Science Basic Research Program of Shaanxi Province, China (No. 2021JQ-170) for their support to this work. The authors are grateful to anonymous reviewers for their effort and time on review of this manuscript.

**Conflicts of Interest:** The authors declare no conflict of interest.

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
