# Peer review of "Soil Aggregate Breakdown with Colloidal Particles Release and Transport in Soil: A Perspective from Column Experiments"

_agriculture, doi:10.3390/agriculture12122155_

Round 1
Reviewer 1 Report
Dear Authors,
the manuscript you have presented meets all the requirements for scientific papers on agricultural topics. You have covered a very interesting topic in agrophysics and soil science. The presented research is original, it can inspire other authors to perform a simple and very important experiment. In my opinion, the manuscript is excellently written. It contains all the necessary elements of a good manuscript, and its most important part, the discussion, is very well written.
Minor remarks: I would very much like to ask you to supplement the paper with photos of the experiment carried out - it can be, for example, a soil profile,
Please paste in the method section a map showing the study area against a map of China.
Calcic Cambisols-are such soils prevalent in China?
Congratulations on your good work!
Reviewer 2 Report
This is the revision of the manuscript number agriculture-2045294 Title: “Soil aggregate breakdown with colloidal particles release and transport in soil: a perspective from column experiments”, proposed by Ms. Daria Li and colleagues for consideration for publication in Agriculture
The manuscript raises an interesting issue regarding the assessment of productive soils. As the basic unit of soil structure, soil aggregates and their stability are closely concerned with the questions of various soil processes and environmental problems. Even so I have many doubts about the study, such as the criteria that were taken to decide the number of soil samples for each with zone the analysis.
Material and methods:
Line 93 How many soil samples were taken and based on what justification these samples were taken.
Results and discussion:
Line 184 Table 1, I think it is necessary to indicate the errors or the statistical study of the data.
Line 251 Figure 2 it is necessary to show the error values to be able to discuss the data
It could be interesting to see the study of more parameters carried out to be able to discuss the data more firmly, the structural stability of the soil is a parameter that could give a lot of information to the study, such as the apparent density of the soil.
General comments:
The author reports that the column experiments were used to investigate the transport properties and soil particles released from the disintegration of model soil aggregates caused by different levels of electrostatic forces. But I do not see in this study any interaction with the mobility of essential nutrients for agriculture.
Reviewer 3 Report
Throughout the work, I found the material and method information regarding the procedures used for soil analysis lacking. Please provide citations of works on which methodology you relied during your analyses or briefly describe your methodology.
